# Use of Pembrolizumab for Treatment of Progressive Multifocal Leukoencephalopathy in People Living with HIV

**DOI:** 10.3390/v14050970

**Published:** 2022-05-05

**Authors:** Carmela Pinnetti, Eleonora Cimini, Alessandra Vergori, Valentina Mazzotta, Germana Grassi, Annalisa Mondi, Federica Forbici, Alessandra Amendola, Susanna Grisetti, Francesco Baldini, Caterina Candela, Rita Casetti, Paolo Campioni, Maria Rosaria Capobianchi, Chiara Agrati, Andrea Antinori

**Affiliations:** 1HIV/AIDS Unit, National Institute for Infectious Diseases, Lazzaro Spallanzani IRCCS, 00149 Rome, Italy; valentina.mazzotta@inmi.it (V.M.); annalisa.mondi@inmi.it (A.M.); susanna.grisetti@inmi.it (S.G.); francesco.baldini@inmi.it (F.B.); caterina.candela@inmi.it (C.C.); andrea.antinori@inmi.it (A.A.); 2Cellular Immunology and Pharmacology Laboratory, National Institute for Infectious Diseases, Lazzaro Spallanzani IRCCS, 00149 Rome, Italy; eleonora.cimini@inmi.it (E.C.); germana.grassi@inmi.it (G.G.); rita.casetti@inmi.it (R.C.); chiara.agrati@inmi.it (C.A.); 3Laboratory of Virology, National Institute for Infectious Diseases, Lazzaro Spallanzani IRCCS, 00149 Rome, Italy; federica.forbici@inmi.it (F.F.); alessandra.amendola@inmi.it (A.A.); maria.capobianchi@inmi.it (M.R.C.); 4Radiology Unit, National Institute for Infectious Diseases Lazzaro Spallanzani IRCCS, 00149 Rome, Italy; paolo.campioni@inmi.it

**Keywords:** PML, advanced HIV-1, opportunistic infection, pembrolizumab, PD-1, JCV

## Abstract

Progressive Multifocal Leukoencephalopathy (PML) is a demyelinating disease occurring in advanced HIV infection, caused by the reactivation of poliomavirus JC (JCV). The use of pembrolizumab for treatment is based on the inhibition of programmed cell death protein 1 (PD-1), potentially improving the anti JCV-specific response. We used pembrolizumab with combined antiretroviral treatment (cART) on a compassionate-use basis. At each administration, clinical evaluation, MRI and laboratory testing, including CD3, CD4, CD8, PD-1 markers, HIV-RNA and JCV-DNA in cerebrospinal fluid (CSF)/plasma pairs, were performed. The JCV-specific T cell response was analysed by Elispot assay. This study included five HIV patients: four male, median age 43 years (29–52), median CD4 and CD8 count 150 (15–158) and 973 (354–1250) cell/mm^3^, respectively; median JCV-DNA and HIV-RNA in CSF/plasma pairs 9.540/1.503 cps/mL and 2.230/619 cp/mL, respectively. Overall, patients received between two and seven doses of pembrolizumab. After treatment, we observed JCV-DNA reduction and PD-1 down-regulation both in CSF and in plasma (high in circulating CD4 and CD8 at baseline), which remained stable at low levels in all patients. Three out of five patients showed stability of clinical picture and neuroimaging, while two others died. More data are needed in order to identify predictors of response to therapy.

## 1. Introduction

Progressive Multifocal Leukoencephalopathy (PML) is a severe demyelinating disease caused by the reactivation of poliomavirus JC (JCV) mainly occurring in immunosuppressive conditions, including HIV infection [1]. The mechanisms by which JCV reactivates and damages the brain parenchyma are currently poorly understood [2]; indeed, both viral and host immunological factors are involved in the pathogenesis of neuronal damage, as the JCV reactivation appears to be more common in patients with CD4 + T cell depletion [3]. Among persons living with HIV (PLWH), the prevalence of PML over time has gradually decreased, suggesting that the immunity restoration induced by an effective combined antiretroviral treatment (cART) is a possible treatment strategy [4,5], in absence of specific anti-JCV therapy. Nevertheless, PML remains a disease burdened by very high morbidity and mortality even in the cART era [6]. The clinical presentation depends on the localization, the extent of demyelination and the rapidity of progression leading to a wide spectrum of severe associated signs and symptoms, including impaired behaviour and cognitive disorders, motor and language deficits, hemianopsia and seizures [7]. The prognosis is poor in most cases, with a 1-year mortality rate of around 30% [8]. Therefore, in recent years, new strategies have been proposed, with the aim of enhancing the specific anti-JCV immune response [9]. Among these new approaches, promising results derive from recent experiences in the use of immune checkpoint-blocking antibodies, including pembrolizumab [10]. Pembrolizumab (PEM) is a monoclonal antibody that inhibits the programmed cell death protein 1 (PD-1), currently approved for the treatment of some cancers. PD-1 expression, a negative immune regulator expressed on activated T cells, is upregulated on CD4+ and CD8+ cells of patients with PML, particularly on JC virus-specific CD8+ T cells [11]. The rationale of its use for treatment of PML is that the inhibition of PD-1 could be potentially associated with better anti JCV-specific response, with consequent JCV clearance and neurological improvement. The aims of our study were to explore JCV-DNA quantitative reduction, PD-1 downregulation and enhanced JCV-specific T cell response after pembrolizumab in people with HIV infection.

## 2. Materials and Methods

We included five HIV-positive subjects with a diagnosis of confirmed PML admitted to the National Institute for Infectious Diseases L. Spallanzani IRCCS in Rome between 1 November 2019 and 15 September 2020. We considered all consecutive HIV-infected patients and contextually ascertained the diagnosis of PML with a high expression of the PD-1 exhaustion marker before treatment. A manual collection review of the medical record was performed, and a clinician-reviewer recorded demographic data and all the clinical data. We employed the Karnofsky performance status scale (KS) to assess patients’ functional impairment (see Table 1). The drug was given on a compassionate-use basis approved by our local Ethics Committee, and all patients provided written informed consent for the administration and for using their data for research purposes (Study Project RetroSNC Ethical approval n.78/2016, 14 June 2016). We used pembrolizumab at a dosage of 2 mg/kg administered intravenously every 4 weeks with cART for treatment of PML. All patients received at least two doses, with a maximum of seven doses. At each pembrolizumab administration, clinical evaluation and magnetic resonance imaging (MRI) with gadolinium-based contrast material were performed; the images of all the patients were assessed, reported and interpreted by one and the same neuroradiologist. Laboratory testing included blood count, blood chemistry tests and immunophenotyping (CD3, CD4, CD8, PD-1 markers) in blood and CSF by flow cytometry. Briefly, whole blood was stained with a cocktail of monoclonal antibodies (specific for the markers described above) for 20 min at 4 °C, then lysed and fixed (lysis buffer, Beckton Dickinson, Firenze, Italy). After fixation, cells were washed and acquired by flow cytometry (Lyric cytometer, BD). Data analysis was performed by BD FACSSuite (v.1.4). Cellular immune response was evaluated by quantifying JCV-specific T cells (specific for VP1 and LT1 JCV proteins) by Elispot assay (Elitech, Berlin, Germany). As a positive control (the evaluation of the immunocompetence), peripheral blood mononuclear cells (PBMCs) were stimulated with phytohemagglutinin (PHA). As a negative control (spontaneous interferon (IFN)-γ release), PBMCs were maintained in a culture medium without specific stimulation. HIV-1 RNA was quantified with the Aptima^®^ HIV-1 Quant Dx assay (Hologic, Inc., San Diego, CA, USA), a totally automated procedure based on real-time TMA (transcription mediated amplification), that utilises a multiplex reaction to amplify, simultaneously and independently, two highly conserved regions of the HIV-1 genome (pol and LTR). The Low Limit of Quantification (LLoQ) of HIV-1 RNA is 30 copies/mL; the Low Limit of Detection (LLoD) of JCV-DNA is 12 copies/mL. JCV-DNA was detected by an in-house RealTime PCR (LightCycler, Roche Diagnostics, Mannheim, Germany), and the targeting of VP1 and JCV load (cps/mL) was executed by using a standard reference curve, with an LoD of 150 cps/mL. This study was supported by Ricerca Corrente Linea 2, Progetto 3 and funded by the Italian Ministry of Health in order to perform immunological tests.

## 3. Results

### 3.1. Clinical Characteristics of Patients

The main characteristics of the five HIV-positive patients included are summarized in Table 1. Briefly, they were four males and one female, with a median age of 43 years old (IQR, 29–52) and a median CD4 and CD8 count of 150 (15–158) and 973 (354–1250) cell/mm^3^, respectively; the median JCV-DNA and HIV-RNA in CSF/plasma pairs were 9540/1503 cps/mL and 2230/619 cp/mL, respectively. At the end of each course of treatment with pembrolizumab, 3 out of 5 patients showed a clinical stability with a progressive reduction in the size of the cerebral lesions on MRI (patient 1, patient 2 and patient 3), but two others died (patient 4 and patient 5). The degree of the clinical disability following PML varied, and the most common deficits observed were visual impairment or hemianopsia, facio-brachial-crural hemiparesis, ataxia, aphasia and cognitive dysfunction, typically associated with psychosis and seizures in two cases. All patients undertook mirtazapine therapy upon the diagnosis of PML. We did not observe any significant side effects during or immediately following pembrolizumab administration in all patients.

Patient 1’s medical history begins in 2014 when he was diagnosed with Castleman Disease and underwent therapy with six cycles every 21 days of R-CHOP (Rituximab, Vincristine, Doxorubicin, Cyclophosphamide and Prednisolone). In 2019, the patient underwent new chemotherapy with four cycles of ABVD (Doxorubicin, Bleomycin, Vinblastine, Decarbazine and Prednisolone) for stage 4 Hodgkin Lymphoma. This treatment was carried out from July 2019 to November 2019 and was suspended after the simultaneous diagnosis of PML. From 2014 to 2019, the patient was lost to follow up and discontinued all therapy, including cART. His CD4 T cell count at LH diagnosis was 160 cells/mmc (16%) and HIV RNA was 1,459,955 copies/mL. Patient 2 has never been compliant with the therapies since her HIV diagnosis, which occurred in 1999, although aware of her HIV/HCV coinfection. At diagnosis of PML, she was off therapy, her CD4 T cell count was 228 cells/mmc, HIV-RNA was 5,950,845 copies/mL and HCV RNA was 678,245 IU/mL, genotype 3. When both patients were re-engaged in care, they restarted with FTC/TDF/DTG, which is still ongoing. After treatment with pembrolizumab, we observed a progressive clinical amelioration with a gradual slow improvement of the radiological picture in both cases. We contextually observed a decline in JCV-DNA in the CSF from 262,800 copies/mL and 9540 copies/mL, respectively, to levels under our cut-off of detectability (<150 cps/mL) within four weeks after pembrolizumab administration. In Patient 3, the diagnosis of HIV infection was obtained during hospitalisation for P. jirovecii pneumonia, treated with prednisolone and CTX IV in June 2020. The cART was started 10 days after the diagnosis with tenofovir alafenamide, emtricitabine and bictegravir, and we observed a rapid decline in plasma HIV RNA (from 222,471 to 619 copies/mL/L) and an increase in CD4 T cells (from 22 cells/mmc, 2.3% to 89 cells/mmc, 6.8%). After about one month from the start of the cART, the patient presented epileptic seizures and a brain MRI showed cerebral lesions compatible with PML, confirmed by the detection in CSF of JCV DNA equal to 183 copies/mL. In a few days we observed a rapid clinical worsening with the appearance of aphasia, behavioural alterations and contrast enhancement, cerebral oedema and mass effect on brain MRI. The CD4 T cell count was 102 cells/mmc (7.2%) and HIV RNA 102 copies/mL. In the suspicion of an IRIS PML, we decided to start therapy with methylprednisolone dosed at 1 g per day for 5 days, followed by an oral prednisone tapered over 1 to 4 weeks, before the first course of pembrolizumab, which we administered after about 15 days. In the following period, the radiological picture showed a further worsening with the appearance of a midline shift, which made it impossible to perform further lumbar punctures for monitoring the JCV-DNA. The clinical case was discussed collectively, and we decided to resume a further course of steroid therapy and postpone the administration of pembrolizumab by approximately two weeks. We administered a total of seven cycles of pembrolizumab complicated by two other episodes of PML IRIS treated with methylprednisolone. Currently, the patient is continuing the same cART. After the initial clinical and radiological worsening, however, there was a gradual, slow functional recovery and reduction in demyelination areas and cerebral oedema on brain MRI. As soon as it was possible to perform lumbar puncture, we documented JCV-DNA below the threshold in each CSF sample. Patients 4 and 5 came to our observation with a rapidly worsening clinical picture characterized by seizures, aphasia and facio-brachio-crural hemiparesis. In both cases, cART was started early, within 7 days of HIV diagnosis (with tenofovir alafenamide, emtricitabine, darunavir/cobicistat plus dolutegravir in patient 4 and tenofovir alafenamide, emtricitabine, bictegravir plus darunavir/cobicistat in patient 5), but the patients showed severe immunosuppression at PML diagnosis (CD4 T cell count was 64/mmc in patient 4 and 15/mmc in patient 5). Although JCV DNA decreased rapidly after the first administration of pembrolizumab (from 338,940 to 26,680 copies/mL in pt4 and from 14,430 to 333 copies/mL in pt5), both patients experienced marked clinical deterioration and too soon died of sepsis due to candida parapsilosis and pneumonia due to MDR pseudomonas aeruginosa, respectively.

### 3.2. Virological and Anti JCV-Specific Response

After treatment, we observed a JCV DNA decline in all patients (median change −0.42, −1.64, −0.09 log). In two patients with stable or slightly improved clinical and radiological pictures, PD-1 expression was high in circulating CD4 and CD8 at baseline, gradually decreased over time and remained stable at low levels in both patients. The expression of PD-1 in CSF was higher than in the peripheral blood, even though it was lower after PEM. All patients experienced an improvement in JCV-specific T cell response after PEM that paralleled PD-1 decrease and JCV-DNA decay in CSF; both patients showed undetectable JCV-DNA in plasma and in CSF. In the clinical case of a 36-year-old patient with PML IRIS, we observed a reduction in the levels of PD1 on circulating CD4 and CD8 T cells, and, when it was possible to perform the lumbar puncture, also in the CSF. Although we documented an initial improvement in the specific response to JCV, after beginning the therapy with dexamethasone we observed a reduction in this response. JCV DNA is persistently undetected in CSF. For the two patients who did not survive, although we documented an improvement in the JCV-specific response, a reduction in PD1 expression and a decline in JCV DNA in both cases, the two patients died from candida sepsis and MDR sepsis and pneumonia after two months of observation, respectively.

## 4. Discussion

According to these preliminary data, JCV-DNA quantitative reduction, PD-1 downregulation and enhanced JCV-specific T-cell response after pembrolizumab treatment were observed in all patients. Nevertheless, the clinical and radiological response was very heterogeneous: two out of five patients maintained a stable, or slightly improved, clinical picture over time, two patients died, and one patient had an immune-reconstitution inflammatory syndrome with clinical improvement. The viro-immunological data and the main radiological findings are shown in Figure 1 and Figure 2, respectively.

The reported five patients had heterogeneous characteristics at PML diagnosis, and, but for one patient (patient 1 restarted the cART four months earlier together with chemotherapy), all were performing antiretroviral therapy for poor adherence or for the lack of knowledge of HIV status and all shared a profound immunosuppression. It is known that the HIV protein Tat promotes transcriptions and the replication of the JCV archetype in vitro, resulting in the potential synergistic effect of HIV on the interruption of JCV latency [12,13]. The JCV reactivation could be associated with multiple JCV quasispecies, with highly variable genetic NCCR rearrangement, that can affect different brain areas due to different cellular and tissue neurotropism [14,15,16].

In all the three patients still alive, we observed, after the first four weeks of cART and pembrolizumab, undetectable HIV RNA and JCV DNA in the CSF. Starting or optimizing cART as soon as possible might limit this synergistic effect on JCV replication, as well as provide a quick restoration of immunity. The HIV and JCV decay both in plasma and in CSF denotes the immune reconstitution, although incomplete. The immune reconstitution remains the most desirable goal for treatment of PML and all treatment strategies aimed at the restoration of cellular immunity, particularly JCV-specific CD8 and CD4 T cells [17]. In our case series, in patient 1 and patient 2, treatment with pembrolizumab may have contributed to strengthening the specific antiviral response characterized by a rapid viral decay and partial improvement of the clinical picture. On the other hand, in the two patients who quickly died, the very advanced diagnosis of the disease and the involvement of more than three brain areas at PML diagnosis, as well as the lack of viral suppression of HIV both in plasma and in CSF, also made the JCV-specific anti-inflammatory response ineffective. A separate discussion is deserved for the patient with PML-IRIS, patient 3. IRIS occurs in about 4–42% of HIV/AIDS patients starting cART [6,18,19] and is the result of an excessive immune response to either pre-existing or latent opportunistic pathogens [20]. PML has been reported to occur within the first weeks to months after initiating cART and has been associated with clinical and radiological worsening [21,22]. In patient 3, we observed, about four weeks after starting cART, a neurological decline and appearance at MRI of contrast enhancement, oedema and mass effect, compatible with a diagnosis of unmasked PML. We started methylprednisolone with partial improvement, but during the pembrolizumab administrations, we observed a fluctuation in the clinical and radiological status, which required the concomitant use of steroids throughout the course of therapy. Systemic corticosteroids have been used empirically in this specifical setting, with reported benefit [23]. Although the clinical improvement after therapy with steroids indirectly confirmed the diagnosis of PML-IRIS in this patient, the prolonged use of methylprednisolone partially affected the specific anti-JCV response. In addition, the low JCV viral load at baseline and the evidence of a relevant inflammatory pattern at MRI probably allowed for a satisfying result in this case. While previous data have shown a survival advantage among PLWH with PML who develop IRIS versus those who do not, it is still debated whether PML-IRIS is an indicator of good clinical outcomes [24]. We can assume that the marked inflammatory response, probably triggered by the almost simultaneous start of cART and checkpoint inhibitors, was indispensable for this patient to obtain a good clinical outcome.

The small number of patients observed does not allow us to draw definitive conclusions regarding the use of pembrolizumab for the treatment of PML in PLWH. Although the strength of this case series is the extensive study of the patients, there are some limitations that should be addressed.

First, our threshold of JCV-DNA is equal to 150 copies/mL; therefore, we are not able to exclude a low-level viremia that may have undergone persistent phenomena of immune activation and inflammation, especially in the case of the patient with PML-IRIS. Second, we have not been able to amplify and sequence JCV’s DNA in order to detect NCCR rearrangements, although we have retained some of the sample aliquots and this is one of our next goals.

Finally, a further limit is represented by the absence of a codified duration of pembrolizumab treatment. In our case series, we treated the three patients who are still alive after seven cycles. Although the drug has been proven to be well tolerated, in patient 1 we had to discontinue its use after myocardial infarction. There are data relating to the cardiotoxicity of pembrolizumab [25,26], but we are not able, in this specific case, to associate the onset of a heart attack with the use of pembrolizumab, as the patient presented with coronary stenosis and other cardiovascular risk factors (including previous chemotherapy, smoking and hypercholesterolemia).

## 5. Conclusions

Although PML does not affect a huge number of people, its course and outcomes are extremely disabling, with death in most cases. The costs, both in terms of care and quality of life, are very high for patients and their families. More data are needed to identify predictors of response to therapy with pembrolizumab in PLWH and PML in order to identify eligible patients with the highest probability of response to this therapy.

## Figures and Tables

**Figure 1 viruses-14-00970-f001:**
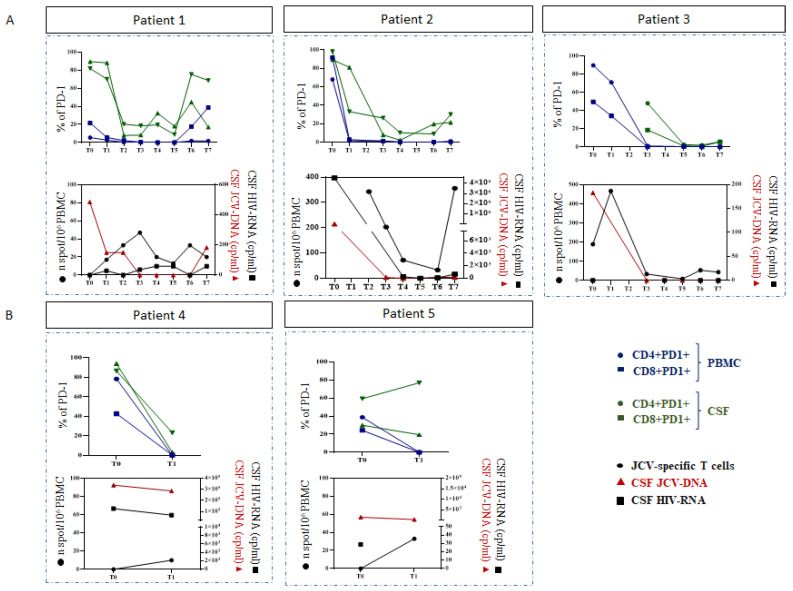
Frequency of circulating and CSF CD4 and CD8 expressing PD-1 was performed by multiparametric flow cytometry; JCV-specific T cell response, JCV-DNA decay and HIV-RNA decay in surviving (**A**) and dead (**B**) patients. JCV-specific T-cell response was performed by Elispot assay; JCV DNA and HIV RNA were tested by Real-Time PCR.

**Figure 2 viruses-14-00970-f002:**
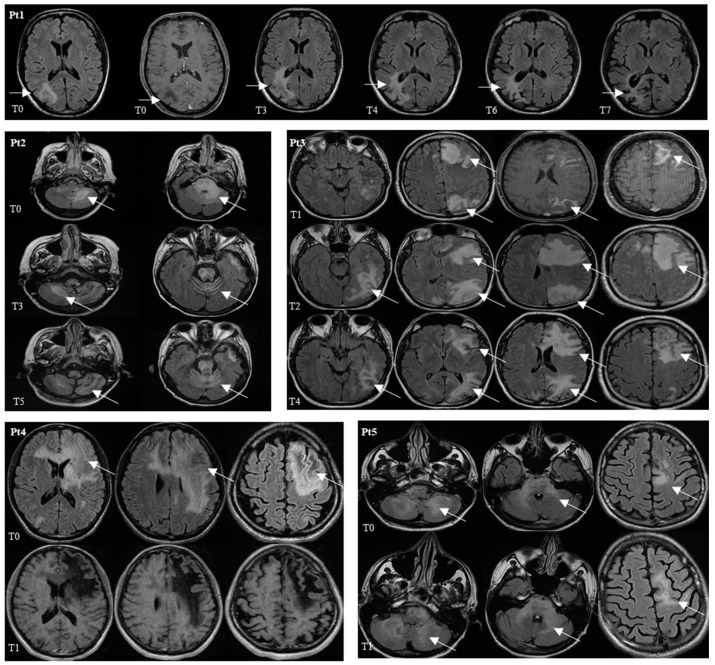
MRI axial T2 Flair of alive Patients (Pt1, Pt2, Pt3) and dead Patients (Pt4 and Pt5) at different timepoints. Pt 1: hyperintensity of white matter in the right parietal region. Pt 2: hyperintensity in the cerebellar hemispheres, brainstem and left temporal pole. Pt 3: hyperintensity in the left frontal and parietal lobe. Pt 4: hyperintensity in white and grey matter of bilateral frontal lobes and genu of corpus callosum. Pt 5: diffuse hyperintensity in cerebellar hemispheres and brainstem. Hyperintensity of white and grey matter of mesial portion of left frontal lobe. In three alive patients the neuroradiological findings showed well-defined inflammatory white matter localizations, including multiple districts. In follow up studies, lesions decreased causing response to therapy. In two dead patients the lesions appeared too extensive with multiple localizations and direct extension into the controlateral hemisphere through corpus callosum fibers and brainstem. In follow up MRI studies, no reductions in lesions extent were observed.

**Table 1 viruses-14-00970-t001:** Characteristics, cART history, viro-immunological assessment before and after pembrolizumab treatment, neurological signs/symptoms and outcomes of patients.

**Patients’ Characteristics**	Pt 1: M, 52 years old, MSM CDC-C3 (Kaposi’s Sarcoma) Previously: R-CHOP for Castelman Disease (2014), ABVD for Hodgkin Lymphoma (2019) KS 80	Pt 2: F, 46 years old, IDU CDC-B3 at diagnosis of HIV Previously: HCV-Ab, arterial hypertension, thyreopathy, psychosis KS 50	Pt3: M, 36 years old, MSM CDC-C3 at diagnosis of HIV (PCP) No other comorbidities KS 70	Pt 4: M, 43 years old, MSM CDC-A2 (PHI) Previously: syphilis KS 60	Pt 5: M, 30 years old, MSM CDC-C3 (PML) Previously: gastritis KS 50
**cART** **History**	2012 to 2016 FTC/TDF/EFV→FTC/TDF/RPV Off therapy until 2019 At T0: FTC/TDF/DTC	Off therapy At T0: FTC/TDF/DTG	Off therapy At T0: BIC/F/TAF	From 2012 to 2016 FTC/TDF/DRV/r→ FTC/TDF/DRV/c→ E/c/FTC/TAF Off therapy since 2016 At T0 (2020): FTC/TAF/DRV/c+DTG	Start 1 month before PML diagnosis with FTC/TAF/BIC+DRV/c
**Viro-Immunological** **Assessment**	**At T0:** CD4 T 282 cells/mmc (15.6%) HIV RNA <30 cps/mL CSF JCV DNA 262,800 cps/mL **At T7:** CD4 T 264 cells/mmc (13%) HIV RNA <30 cps/mL CSF JCV DNA 20 cps/mL	**At T0:** CD4 T 158 cells/mmc (11.2%) HIV RNA 5,950,845 cps/mL CSF JCV DNA 9540 cps/mL **At T5:** CD4 T 276 cells/mmc (17.5%) HIV RNA <30 cps/mL CSF JCV DNA 33 cps/mL	**At T0:** CD4 T 89 cells/mmc (6.8%) HIV RNA 619 cps/mL CSF JCV DNA 183 cps/mL **At T5:** CD4 T 55 cells/mmc (6%) HIV RNA <30 cps/mL CSF JCV DNA 43 cps/mL	**At T0:** CD4 T 64 cells/mmc (13.2%) HIV RNA 28,344 cps/mL CSF JCV DNA 338,910 cps/mL **At T2:** CD4 T 75 cells/mmc (11.5%), HIV RNA <30 cps/mL CSF JCV DNA 10 cps/mL	**At T0:** CD4 T 15 cells/mmc (2.2%), HIV RNA 351 cps/mL CSF JCV DNA 14,430 cps/mL **At T2:** CD4 T 35 cells/mmc (4.5%) HIV RNA 170 cps/mL CSF JCV DNA 33 cps/mL
**Neurological Signs/** **Symptoms**	Left hemianopia and loss of vision	Ataxia, loss of vision, left dysmetria and psychosis	Aphasia, confusion and comitial crisis	Aphasia and faciobrachial crural hemiparesis	Cognitive impairment, behavioral disturbances, progressive facio-brachial-crural hemiparesis, aphasia
**Clinical** **Outcomes**	Clinical and radiological improvement	Lack of evolution Clinical and radiological stability	Clinical and radiological improvement	Death due to sepsis	Death due to sepsis

## Data Availability

Data sharing will be available upon reasonable request.

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
