# Peer review of "Use of Pembrolizumab for Treatment of Progressive Multifocal Leukoencephalopathy in People Living with HIV"

_viruses, 2022, doi:10.3390/v14050970_

Round 1

Reviewer 1 Report

In the present manuscript, Pinnetti and collaborators explore the use of Pembrolizumab as for the treatment of Leukoencephalopathy (PML) caused by JC-virus reactivation in HIV+ individuals. The article provides interesting data about the reduction of the JC viral load associated to improved immune reconstitution together with recovery from disease progression.

My major concern regarding the manuscript is the low number of patients. Authors present the data of 5 individuals and 2 out of them died. Together with the heterogeneity of the individuals, it is difficult to establish consistent conclusion apart from preliminary data or proof-of-concept. In that sense, the data is valuable as demonstration of how the use of checkpoint inhibitors to treat JC-derived malignancies as already proposed in Cortese et al., Nat Rev Neurol. 2021; Bernard-Valnet, Ann Neurol. 2021. However, due to the low number of patients (3 in the end), authors should consider to improve the manuscript with further data, as for example in vitro determinations to mechanistically explain the role of PD1 therapy, controls with/without PD1 treatment and combination or not with anti-HI drugs, in the context of JC virus replication (Barth et al., Viruses. 2016 Oct; 8(10): 292).

Minor concerns:

  • Flow cytometry is poorly explained.
  • Figure 1. Add HIV viral load if possible to graphs.
  • Figure 2. It is out of my expertise but I wonder if any kind of quantification/correlation could be established between lesions/images/damage reduction/JC viral load?

Author Response

Comments and Suggestions for Authors

In the present manuscript, Pinnetti and collaborators explore the use of Pembrolizumab as for the treatment of Leukoencephalopathy (PML) caused by JC-virus reactivation in HIV+ individuals. The article provides interesting data about the reduction of the JC viral load associated to improved immune reconstitution together with recovery from disease progression.

My major concern regarding the manuscript is the low number of patients. Authors present the data of 5 individuals and 2 out of them died. Together with the heterogeneity of the individuals, it is difficult to establish consistent conclusion apart from preliminary data or proof-of-concept. In that sense, the data is valuable as demonstration of how the use of checkpoint inhibitors to treat JC-derived malignancies as already proposed in Cortese et al., Nat Rev Neurol. 2021; Bernard-Valnet, Ann Neurol. 2021. However, due to the low number of patients (3 in the end), authors should consider to improve the manuscript with further data, as for example in vitro determinations to mechanistically explain the role of PD1 therapy, controls with/without PD1 treatment and combination or not with anti-HI drugs, in the context of JC virus replication (Barth et al., Viruses. 2016 Oct; 8(10): 292).

The small number of patients observed does not allow us to draw definitive conclusions regarding the efficacy of pembrolizumab as treatment of PML in PLWH. The strength of this case series is the extensive and comprehensive study of the patients. Randomized clinical trials that allow the comparison between patients treated or not with Pembrolizumab (alone or in combination with other drugs) are certainly desirable. PML rarely occurs among people with HIV, especially after the introduction of cART and the outcomes related to this disease are often serious and fatal. Therefore, having a potential drug available for its treatment could be very relevant.

Shortly, our data will be aggregated to a broader international cohort that also involves patients with PML suffering from other diseases (including multiple sclerosis, onco-hematological diseases and primary immunodeficiencies), which will give more information on the clinical impact of the use of Pem for the treatment of PML

Aims of our study was to explore JCV-DNA quantitative reduction, PD-1 down regulation and enhanced JCV-specific T-cell response after pembrolizumab in people with HIV infection.

Minor concerns:

  • Flow cytometry is poorly explained.

As suggested, we improved the description of the flow cytometry and added a sentence in the methods Section (lines 76-80). 

  • Figure 1. Add HIV viral load if possible to graphs.

As suggested, we added in Figure 1 the kinetic of HIV-RNA in CSF

  • Figure 2. It is out of my expertise but I wonder if any kind of quantification/correlation could be established between lesions/images/damage reduction/JC viral load?

 At the present time we do not have enough data to be able to suppose a relationship between JCV-DNA decay and image changes on brain MRI. We cannot draw definitive conclusions, but we can assume that an increased immunological response at the CNS level by check point inhibitors could be associated with a different neurological imaging.

Reviewer 2 Report

This is an interesting and significant case series investigating the anti-PD-1, pembrolizumab in the treatment of PML in HIV patients. Even if this kind of work is remarkable in the field, it is important that the information is well structured and written, for this, I believe, it’s suitable for publishing in “Viruses”.

However, some minor points need to be addressed:

Please omit the acronyms/abbreviations in the article title, abbreviations that are used in your paper should be introduced on the first usage in the abstract, and again if/when they appear in the main body of the document.

Please write down clearly what are the main objectives of the study.

I suggest mentioning the eligibility criteria required for treatment with pembrolizumab in the study setting especially the CD4 count

Please formulate the results in paragraphs to help the readers follow the finding.

It would be interesting as well to further clarify the stated IRIS and immune-related adverse events and mention any Pembrolizumab adverse effects.

Author Response

Comments and Suggestions for Authors

This is an interesting and significant case series investigating the anti-PD-1, pembrolizumab in the treatment of PML in HIV patients. Even if this kind of work is remarkable in the field, it is important that the information is well structured and written, for this, I believe, it’s suitable for publishing in “Viruses”.

However, some minor points need to be addressed:

Please omit the acronyms/abbreviations in the article title, abbreviations that are used in your paper should be introduced on the first usage in the abstract, and again if/when they appear in the main body of the document.

Thank you, acronyms/abbreviations have been removed form the title

Please write down clearly what are the main objectives of the study.

The main objectives of the study have been reported in the main text (lines 55-57)

I suggest mentioning the eligibility criteria required for treatment with pembrolizumab in the study setting especially the CD4 count

As suggested, we included a sentence for better define the population included (lines 61-63). Future studies are needful to identify predictors of response to therapy.

Please formulate the results in paragraphs to help the readers follow the finding.

The "Results" section has been split into further sub-sections: Clinical characteristics of patients and Virological and anti JCV-specific response

It would be interesting as well to further clarify the stated IRIS and immune-related adverse events and mention any Pembrolizumab adverse effects

We have extensively described the clinical case of patient 3 (IRIS PML lines 127-151). However, as suggested, we have added a summary sentence to the discussion to better clarify how much, in our opinion, the inflammatory response may have conditioned the clinical outcome (lines 242-245). We did not observe any significant side effects during or immediately following pembrolizumab administration in all patients. We have clarified the concept in lines 108-109.

Reviewer 3 Report

In this manuscript, Andrea Antinori and his colleagues reported the effect of Pembrolizumab on PML treatment in HIV patients. They used Pembrolizumab together with cART to treat PML in PLWH and found JCV-DNA reduction, PD-1 down-regulation, and improved JCV-specific T cell response in all five patients. Although the limited patient samples displayed heterogeneous clinical outcomes and caused an unclear conclusion on the effect of Pembrolizumab treatment, the detailed data provide a reference for further studies. Overall, the results of this manuscript were clearly presented, and the discussion was adequate and objective. Therefore, I have only a few comments below.

Minor comments:

  1. Please also show the full name of CSF, LoQ and LoD when they were first used.
  2. Line 89, in the sentence “they were 4 males and 1 woman”, a woman should be replaced with a female.
  3. In Fig. 2, although the description of the focus was clear in the legend, if the authors used arrows to indicate them, it would be more distinct to readers.

Author Response

In this manuscript, Andrea Antinori and his colleagues reported the effect of Pembrolizumab on PML treatment in HIV patients. They used Pembrolizumab together with cART to treat PML in PLWH and found JCV-DNA reduction, PD-1 down-regulation, and improved JCV-specific T cell response in all five patients. Although the limited patient samples displayed heterogeneous clinical outcomes and caused an unclear conclusion on the effect of Pembrolizumab treatment, the detailed data provide a reference for further studies. Overall, the results of this manuscript were clearly presented, and the discussion was adequate and objective. Therefore, I have only a few comments below.

Minor comments:

  1. Please also show the full name of CSF, LoQ and LoD when they were first used.

We have now shown the full names of anu acronyms/abbreviations (see Line 19; 88-90)

  1. Line 89, in the sentence “they were 4 males and 1 woman”, a woman should be replaced with a female.

Gender rewording has been done (line 98).

  1. In Fig. 2, although the description of the focus was clear in the legend, if the authors used arrows to indicate them, it would be more distinct to readers

An arrow pointing the lesion have been inserted

Round 2

Reviewer 1 Report

Authors have fully addressed the comments of the previous review. No other comments or concerns.